# Analysis of Antimicrobial Resistance Genes (ARGs) in Enterobacterales and *A. baumannii* Clinical Strains Colonizing a Single Italian Patient

**DOI:** 10.3390/antibiotics12030439

**Published:** 2023-02-23

**Authors:** Alessandra Piccirilli, Elisa Meroni, Carola Mauri, Mariagrazia Perilli, Sabrina Cherubini, Arianna Pompilio, Francesco Luzzaro, Luigi Principe

**Affiliations:** 1Department of Biotechnological and Applied Clinical Sciences, University of L’Aquila, 67100 L’Aquila, Italy; 2Clinical Microbiology and Virology Unit, “A. Manzoni” Hospital, 23900 Lecco, Italy; 3Department of Medical, Oral and Biotechnological Sciences, “G. d’Annunzio” University of Chieti-Pescara, 66100 Chieti, Italy; 4Center for Advanced Studies and Technology (CAST), “G. d’Annunzio” University of Chieti-Pescara, 66100 Chieti, Italy; 5Clinical Pathology and Microbiology Unit, “S. Giovanni di Dio” Hospital, 88900 Crotone, Italy

**Keywords:** NDM-1, Enterobacterales, *A. baumannii*, antimicrobial resistance

## Abstract

The dramatic increase in infections caused by critically multidrug-resistant bacteria is a global health concern. In this study, we characterized the antimicrobial resistance genes (ARGs) of *K. pneumoniae*, *P. mirabilis*, *E. cloacae* and *A. baumannii* isolated from both surgical wound and rectal swab of a single Italian patient. Bacterial identification was performed by MALDI-TOF MS and the antimicrobial susceptibility was carried out by Vitek 2 system. The characterization of ARGs was performed using next-generation sequencing (NGS) methodology (MiSeq Illumina apparatus). *K. pneumoniae*, *P. mirabilis* and *E. cloacae* were resistant to most β-lactams and β-lactam/β-lactamases inhibitor combinations. *A. baumannii* strain was susceptible only to colistin. The presence of plasmids (IncN, IncR, IncFIB, ColRNAI and Col (MGD2)) was detected in all Enterobacterales but not in *A. baumannii* strain. The IncN plasmid and *bla*_NDM-1_ gene were found in *K. pneumoniae*, *P. mirabilis* and *E. cloacae*, suggesting a possible transfer of this gene among the three clinical species. Conjugation experiments were performed using *K. pneumoniae* (1 isolate), *P. mirabilis* (2 isolates) and *E. cloacae* (2 isolates) as donors and *E. coli* J53 as a recipient. The *bla*_NDM-1_ gene was identified by PCR analysis in all transconjugants obtained. The presence of four different bacterial species harboring resistance genes to different classes of antibiotics in a single patient substantially reduced the therapeutic options.

## 1. Introduction

Antimicrobial resistance represents one of the most serious global public health issues [1]. Several microorganisms known as ESKAPE (*Enterococcus faecium*, *Staphylococcus aureus*, *Klebsiella pneumoniae*, *Acinetobacter baumannii*, *Pseudomonas aeruginosa* and *Enterobacter* spp.) have emerged as globally critical multidrug-resistant (MDR) pathogens requiring continues monitoring and development of new drugs [2,3]. To date, only few drugs in development are potentially active against ESKAPE pathogens [4]. Among these, the Gram-negative *P. aeruginosa*, *K. pneumoniae* and other Enterobacterales are of great concern for their high level of resistance to most antibiotics and, in particular, to carbapenems, often considered as our last line of defense against critical pathogens [5,6,7]. In the last decade, infections caused by carbapenemase-producing Enterobacterales (CPE) have dramatically increased [6,7,8,9,10,11]. In particular, the KPC and NDM variants are the most widespread carbapenemases in clinical strains with a wide range of infections [12,13,14,15].

This study reports a detailed characterization of the antimicrobial resistance genes (ARGs) in *P. mirabilis*, *K. pneumoniae*, *E. cloacae* and *A. baumannii*, isolated from surgical wound (SW) and rectal swab (RS) of a particular hospitalized patient.

## 2. Results

### 2.1. Antimicrobial Susceptibility

Overall, six clinical isolates obtained from SW and RS were analyzed against a large panel of antibiotics. As given in Table 1, the *E. cloacae* and *K. pneumoniae* isolates showed resistance to carbapenems (ertapenem, imipenem and meropenem), cephalosporins (cefotaxime, ceftazidime and cefepime), amoxicillin–clavulanic acid, piperacillin–tazobactam, ceftolozane–tazobactam and ceftazidime–avibactam. However, *E. cloacae* and *K. pneumoniae* isolates were susceptible to amikacin, gentamycin and colistin. A similar susceptibility profile was found for *P. mirabilis* isolates. The *A. baumannii* strain was susceptible only to colistin.

### 2.2. Multi-Locus Sequence Typing (MLST) and Plasmid Multi-Locus Sequence Typing (pMLST)

The draft genome of the six isolates recovered for this study was obtained using MiSeq Illumina platform. The sequence analysis revealed that the total number of sequenced nucleotides was 4.6–4.8 Mb for *E. cloacae* isolates, 3.7–4.3 Mb for *P. mirabilis* isolates, 5.7 Mb for *K. pneumoniae* and 3.7 Mb for *A. baumannii* (Table 2). The MLST of *K. pneumoniae*, *E. cloacae* and *A. baumannii* indicates that these strains belonged to the lineages ST4587, ST45 and ST2, respectively. To date, molecular characteristics and genetic relationships among *Proteus* spp. have not been elucidated, and for this reason, the MLST analysis of *P. mirabilis* was not launched.

### 2.3. ARGs and Mobile Genetic Elements

With the exception of *A. baumannii* SW, plasmid replicons have been found in all strains (Table 2). Of note, *P. mirabilis* isolates carried IncN and IncQ1 incompatibility plasmids. Both *E. cloacae* RS and *E. cloacae* SW isolates harbored IncN, IncFIB (pECLA) and IncFII (pECLA) plasmids, but *E. cloacae* RS had, in addition, ColRNAI. *K. pneumoniae* RS carried the IncN, IncR, IncFIB (K), Col (MGD2) and ColRNAI plasmids. Overall, *E. cloacae* RS, *E. cloacae* SW, *P. mirabilis* RS, *P. mirabilis* SW and *K. pneumoniae* RS shared the same IncN plasmid belonging to the ST7 pMLST scheme. Different insertion sequences (ISs) were identified in the six isolates analyzed. Both *E. cloacae* RS and *E. cloacae* SW carried transposon Tn*2*, IS*6100* (IS*6* family), IS*Kpn8* (IS*3* family) and IS*Sen3* (IS*21* family). Nevertheless, in *E. cloacae* RS, IS*Sen4* (IS*3* family), IS*26* (IS*6* family) and IS*Kpn19* (IS*Kra4*) were also detected. IS*Kpn19* and IS*6100* were identified in *P. mirabilis*, *E. cloacae* and *K. pneumoniae* isolates (Table 2). In *A. baumannii*, IS*Aba24* (IS*66* family), IS*Aba26* (IS*256* family) and IS*26* sequence insertions were found. Table 2 displayed a detailed analysis of the resistome of the six clinical strains. Both *P. mirabilis* SW and RS isolates showed the same resistance genes: (a) *bla*_NDM-1_ and *bla*_TEM-1B_, β-lactams resistance genes; (b) *qnrS1*, a plasmid mediating resistance to fluoroquinolones; (c) *aadA1* that confers resistance to aminoglycosides (streptomycin and spectinomycin); (d) *strA-strB* chromosomal gene cluster conferring resistance to streptomycin; (e) *sul*2, *tet* (J) and *catA1*, conferring resistance to sulphonamides, tetracycline and chloramphenicol, respectively. The *E. cloacae* SW and RS isolates harbored *strB*, *strA*, *sul2*, *dfrA14*, *qnrS1*, *catA2*, *bla*_NDM-1_, *bla*_TEM-1B_ and *bla*_ACT-15_, whereas *K. pneumoniae* RS had *qnrS1*, *oqxB*, *dfrA14*, *fosA*, *bla*_NDM-1_ and *bla*_LEN-22_. *A. baumannii* harbored *bla*_OXA-23_, *bla*_ADC-25_, *bla*_OXA-66_, *armA*, *strA*, *strB*, *mph* (*E*), *msr* (*E*), *sul2* and *tetB* genes.

### 2.4. Conjugation Experiments and PCR Assays

Conjugation experiments were performed using *E. coli* J53 strain as a recipient and *P. mirabilis* (RS and SW), *E. cloacae* (RS and SW) and *K. pneumoniae* RS as donors. Conjugational transfer of meropenem resistance was ascertained in all three systems, and the presence of *bla*_NDM-1_ was confirmed by PCR in all transconjugants obtained.

## 3. Discussion

Here, we described the molecular characterization of ARGs of *K. pneumoniae*, *E. cloacae*, *P. mirabilis* and *A. baumannii* clinical strains isolated from different sample sites of a single hospitalized patient. The case was not epidemiologically related to other hospitalized patients, and no information was available about the stay of the patient in LTCF. The characterization of ARGs was performed by NGS analyzing the resistome of all strains. On the basis of the draft genome analysis, we have noted that the IncN plasmid was found in all Enterobacterales analyzed. The IncN belongs to a broad-host-range plasmids with a size of 30 to 70 Kb that, often, carry a great variety of resistance genes, including *bla*_CTX-M_, *bla*_VIM_ and *bla*_NDM_ [16,17,18,19]. In this study, IncN was simultaneously present with IncQ1 in *P. mirabilis*, with IncFIB and ColRNAI in *E. cloacae* and with IncR in *K. pneumoniae*. The IncQ1 belongs to the MOBQ group with a medium–small size (8–14 Kb) and, often, they carry *sul-strA-strB* gene cluster [18]. The IncR (40–160 Kb) is a mobilizable plasmid frequently cointegrated with IncN plasmid [19], the same cluster found in *P. mirabilis* and *E. cloacae* (this study). The β-lactams resistance genes found in *K. pneumoniae*, *P. mirabilis* and *E. cloacae* were *bla*_NDM-1_, *bla*_ACT-15_, *bla*_LEN-22_ and *bla*_TEM-1B_. However, only *bla*_NDM-1_ was found in all Enterobacterales, and for this reason, we have supposed that IncN plasmid harbored *bla*_NDM-1_ gene. On the basis of conjugation experiments, we have presumed the circulation of *bla*_NDM-1_ gene via IncN plasmid among *K. pneumoniae*, *P. mirabilis* and *E. cloacae* isolated from the single patient. The wide distribution of *bla*_NDM-1_ and its natural variants among clinical and community-acquired Enterobacterales is related to the fact that they can be carried by different plasmid types (IncA/C, IncF, IncL/M or untypable) that are readily self-transmissible by conjugation [20]. The *bla*_NDM_ promiscuity is related to its high mobilization capacity into plasmids or chromosomes [21]. Patients simultaneously infected and/or colonized with multiple species of CPE are more frequently observed [22,23,24]. Several cases of interspecies exchange of identical *bla*_KPC-_, *bla*_OXA-48−_ and *bla*_NDM-1_-carrying plasmids have been described [22,23,24,25,26]. In particular, those involving the *bla*_NDM-1_ were mainly related to the horizontal spread of broad-host-range IncC plasmids (formerly, IncA/C2) [26]. Invasive infections by MBL-producing Enterobacterales are associated with high mortality rates (>30%), especially, in the hospital setting when critically ill patients are involved [27,28]. The spread of CPE is significantly increasing in healthcare settings and, also, in long-term care facilities (LTCFs) [15,16,17]. The draft genome analysis of *A. baumannii* strain, analyzed in this study, exhibited the presence of *bla*_OXA-23_, *bla*_OXA-66_ and *bla*_ADC-25_ genes. It is very common to find the simultaneous presence of *bla*_OXA-23_ and *bla*_OXA-66_ in the genome of carbapenem-resistant *A. baumannii* strains [29,30]. The OXA-66 is an OXA-51-like enzyme, intrinsically overexpressed in *A. baumannii* strains, which is able to confer high resistance to carbapenems [31]. The *bla*_ADC_ genes are also chromosomally encoded in *A. baumannii* strains [32]. The presence of IS*Aba24* and IS*Aba26* upstream the *bla*_OXA_ genes indicates the plausible presence of a strong promoter that drives expression of the downstream genes and facilitates the transferability of resistance determinants [33,34]. In particular, the LTCFs represent an important ARGs’ reservoir in older resident people who are more vulnerable to bacterial infections due to multiple chronic illnesses.

## 4. Materials and Methods

### 4.1. Clinical Case Description

In September 2020, an 88-year-old woman was admitted to the Emergency Department of the A. Manzoni Hospital (Lecco, Italy) following an accidental fall. The X-ray revealed a displaced fracture of the left femur involving the lesser trochanter. The patient’s medical history revealed previous fractures of the same femur, presumed autoimmune liver disease, diabetes, obesity and lower limb polyneuropathy. No recent hospitalization was reported. Four days after admission, the patient underwent surgery after washing the fracture site. One week after surgery, the patient was discharged from the Orthopedics unit to an LTCF; but a few days later, she was newly admitted to the General Medicine of the A. Manzoni Hospital, showing hypotension and diffuse icterus. Based on the hospital protocol for patients coming from LTCF, a rectal swab for CPE screening was performed, whereas an antimicrobial treatment was empirically initiated with amoxicillin/clavulanic acid (0.625 g, tid). Laboratory data showed increased values of lipases, bilirubin, AST and ALT enzymes, thus suggesting hepatic dysfunction. Based on appropriate imaging, acute cholecystitis with gallbladder hydrops was diagnosed and empiric therapy was then changed to piperacillin/tazobactam (4.5 g, tid) and gentamicin (240 mg, once a daily). Bacterial isolates were recovered from MacConckey agar (bioMérieux, Marcy l’Etoile, France), after an 18–22 h incubation period in aerobic conditions (37 °C) in the context of a laboratory clinical routine. In particular for rectal swabs, bacterial isolates resistant to carbapenems were also recovered from chromogenic Brilliance CRE agar (Thermo Fisher Scientific, Waltham, MA, USA). Cultures from SW were positive for *E. cloacae* complex SW and *P. mirabilis* SW, both producing an NDM-type carbapenemase. The RS performed at the same time showed positive for *E. cloacae* complex RS, *P. mirabilis* RS and *K. pneumoniae* isolates, all of them producing an NDM-type enzyme. Subsequently, a carbapenem-resistant *A. baumannii* SW was also isolated from the wound. The patient was discharged after 45 days in good health conditions, and a home care regimen was assessed.

### 4.2. Strains Identification and Antibiotic Susceptibility Testing

The bacterial identification was performed by MALDI-TOF mass spectrometry (Vitek MS, bioMérieux, Marcy l’Étoile, France). The antimicrobial susceptibility was determined using both the Vitek 2 system (bioMérieux, Marcy l’Étoile, France) and the Sensititre™ Gram Negative Panel (ThermoFisher, Waltham, MA, USA). Susceptibility results were interpreted according to current EUCAST criteria. Carbapenemase production was first assessed using phenotypic methods, including an immunochromatographic assay (RESIST-4 O.K.N.V., Coris Bio-Concept, Gembloux, Belgium) and specific inhibitor disks (KPC+MBL Confirm ID Kit, Rosco Diagnostica), Appendix A. The clinical strains analyzed in this study were from the surgical wound (*E. cloacae* SW, *P. mirabilis* SW, *A. baumannii* SW) and rectal swab (*E. cloacae* RS, *P. mirabilis* RS, *K. pneumoniae* RS) samples.

### 4.3. Resistome Analysis

Total DNA of *P. mirabilis* (2 isolates), *E. cloacae* (2 isolates), *K. pneumoniae* (1 isolate) and *A. baumannii* (1 isolate) was extracted using a modified protocol, as previously reported [16,17]. Libraries were sequenced using the Illumina MiSeq system by 2 × 300 paired-end approach (Illumina, San Diego, CA, USA) [16,17]. Paired-end reads were assembled using Velvet (v.1.2.10) [35]. The resistome and plasmidome were analyzed using ResFinder 4.1 (available online: https://cge.cbs.dtu.dk/services/ResFinder/ (accessed on 14 January 2022)) and PlasmidFinder 2.1 (available online: https://cge.cbs.dtu.dk/services/PlasmidFinder/ (accessed on 16 January 2022)), respectively [36,37]. MobileElementFinder was used to identify mobile genetic elements and their relation to ARGs [38]. The Pasteur multi-locus sequence typing (MLST) scheme was used to assign the ST (available online: https://bigsdb.pasteur.fr/index.html (accessed on 16 January 2022)) [39].

### 4.4. Conjugation Assays

Conjugation experiments were performed using *Escherichia coli* J53 (rifampicin-resistant strain) as a recipient and *P. mirabilis* (RS and SW), *E. cloacae* (RS and SW) and *K. pneumoniae* RS strains as donors. Transconjugants were selected on Luria–Bertani (LB) agar plates supplemented with 300 mg/L rifampicin and 2 mg/L meropenem or 100 mg/L ampicillin. The detection sensitivity of the assay was ≥5 × 10–7 transconjugants per recipient.

### 4.5. PCR Experiments

One colony of each transconjugant was picked, dissolved in 100 µL of sterile H_2_O and boiled at 100 °C. The mixture was harvested at 14,000× *g*, and 2 µL of supernatant was used for PCR experiments with specific NDM-1 primers. Primers and PCR conditions were from our previous study [40].

## 5. Conclusions

In this study, we have characterized the ARGs from four different bacterial species isolated from a single patient, who was admitted to the Orthopedics unit of the A. Manzoni Hospital. The patient was then discharged from the Orthopedics to an LTCF. After a few days, the patient was newly admitted to the General Medicine of the same hospital for different problems. This represents a clear example of bacterial pathogens spreading from community to hospital settings and vice versa. Unfortunately, we have no information about the epidemiological situation of the LTCF that accommodated the patient. Results obtained in this study showed the persistence of *bla*_NDM-1_ in three different Enterobacterales species isolated from a single patient. NDM producers are not commonly related to the Italian epidemiologic context, but they are emerging and increasingly reported [41,42]. Moreover, infections caused by pathogens harboring ARGs that confer resistance to different classes of antibiotics substantially reduced the therapeutic options, especially, when bacteria harbored MBLs.

## Figures and Tables

**Table 1 antibiotics-12-00439-t001:** Susceptibility profile of isolates obtained from surgical wound (SW) and rectal swab (RS).

	*E. cloacae* SW *E. cloacae* RS		*P. mirabilis* SW *P. mirabilis* RS		*A. baumannii*		*K. pneumoniae* RS	
Antimicrobial Agent	MIC (mg/L)	Interpretation	MIC (mg/L)	Interpretation	MIC (mg/L)	Interpretation	MIC (mg/L)	Interpretation
Amoxicillin/clavulanic acid	>16	R	>16	R	>16	R	>16	R
Piperacillin/tazobactam	>64	R	>32	R	>64	R	>64	R
Cefepime	>16	R	4	I	16	R	8	R
Cefotaxime	>32	R	16	R	>32	R	16	R
Ceftazidime	>32	R	>32	R	>32	R	>32	R
Ceftazidime/avibactam	>16	R	>16	R	>16	R	>16	R
Ceftolozane/tazobactam	>32	R	>32	R	>32	R	>32	R
Ciprofloxacin	>2	R	0.5	I	>2	R	>2	R
Ertapenem	>4	R	>4	R	>4	R	>4	R
Imipenem	>8	R	>8	R	>8	R	>8	R
Meropenem	>8	R	>8	R	>8	R	>8	R
Amikacin	≤1	S	4	S	>32	R	≤1	S
Gentamycin	≤1	S	≤1	S	>8	R	≤1	S
Colistin	0.5	S	<4	R	≤0.5	S	≤0.5	S

Minimum Inhibitory Concentration (MIC) interpretation: R, resistant; I, intermediate; S, susceptible.

**Table 2 antibiotics-12-00439-t002:** Resistome of the six clinical strains isolated from surgical wound (SW) and rectal swab (RS) of a single patient.

Strains	Genome Size (bp)	MLST (Pasteur)	Plasmid Replicons/pMLST	Mobile Genetic Elements	Β-lactams Resistant Genes	Other ARGs
*Proteus mirabilis* RS	4.342.694	none	IncN, IncQ1/ IncN: ST7	IS*Kpn19*, IS*6100*, IS*Vsa5* (IS10R), IS*26*	*bla*_NDM-1_, *bla*_TEM-1B_	*aadA1*, *strB*, *strA*, *sul2*, *dfrA1*, *dfrA14*, *qnrS1*, *tet(J)*, *catA1*
*Proteus mirabilis* SW	3.796.792	none	IncN, IncQ1/ IncN: ST7	IS*Kpn19*, IS*26*, IS*6100*	*bla*_NDM-1_, *bla*_TEM-1B_	*aadA1*, *strB*, *strA*, *sul2*, *dfrA1*, *dfrA14*, *qnrS1*, *tet(J)*, *catA1*
*Enterobacter cloacae* RS	4.617.198	ST45	IncN, IncFIB(pECLA), IncFII(pECLA), ColRNAI/ IncN: ST7	Tn*2*, IS*Kpn19*, IS*26*, IS*6100*, IS*Sen4* (IS3, Group IS407), IS*Sen3*(Family IS21), IS*Kpn8* (Family IS3)	*bla*_NDM-1_, *bla*_TEM-1B,_ *bla*_ACT-15_	*strB*, *strA*, *sul2*, *dfrA14*, *qnrS1*, *catA2*
*Enterobacter cloacae* SW	4.781.639	ST45	IncN, IncFIB(pECLA), IncFII(pECLA)/ IncN: ST7	Tn*2* IS*Sen3* (Family IS21) IS*Kpn8* (Family IS3) IS*6100*	*bla*_NDM-1_, *bla*_TEM-1B,_ *bla*_ACT-15_	*strB*, *strA*, *sul2*, *dfrA14*, *qnrS1*, *catA2*
*Klebsiella pneumoniae* RS	5.757.187	ST4587	IncN, IncR, Col(MGD2), IncFIB(K), ColRNAI/IncN: ST7-like	IS*Kpn19*, IS*Kpn21*, IS*6100*, IS*5075* (Family IS110)	*bla*_NDM-1_, *bla*_LEN-22_	*qnrS1*, *oqxB*, *dfrA14*, *fosA*
*Acinetobacter baumannii* SW	3.737.728	ST2	none	IS*Aba24* (Family IS66) IS*Aba26* (Family IS256) IS*26*	*bla*_OXA-23_, *bla*_ADC-25_, *bla*_OXA-66_	*armA*, *strA*, *strB*, *mph(E)*, *msr(E)*, *sul2*, *tetB*

MLST, Multi-Locus Sequence Typing; pMLST, plasmid Multi-Locus Sequence Typing; ARGs, Antimicrobial Resistance Genes.

## Data Availability

Not applicable.

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
