# Peer review of "Analysis of Antimicrobial Resistance Genes (ARGs) in Enterobacterales and A. baumannii Clinical Strains Colonizing a Single Italian Patient"

_antibiotics, 2023, doi:10.3390/antibiotics12030439_

Round 1
Reviewer 1 Report
the article is very interresting and adds up needed informtion. please, reduce the article and present it as a full letter or corresponcy fowlloing the following recommendations.
please follow up and add the following article which present very important regional information.
Colistin-resistant carbapenemase-producing isolates among Klebsiella spp. and Acinetobacter baumannii in Tripoli, Libya - ScienceDirect
Line 2: i would prefer "analysis" instead of "dissemination" so please modify the title to follow the format over recommendation into letter instead of research article.
line 19:
review the following sentence and make it general as or following to the following: "The dramatic increase of infections caused by critically multidrug resistant bacteria is a global health concern"
line 38: modify "Public Health issue" and no need to capitalize it.
line 41: delete "for which new antimicrobial development is urgently needed" and emerge it with the following short sentence into a good fluent sentence like or similar to:
"Various bacterial organisms known as ESKAPE (Enterococcus faecium, Staphylococcus aureus, Klebsiella pneumoniae, Acinetobacter baumannii, Pseudomonas aeruginosa and Enterobacter spp.) have emerged as globally critical MDR pathogens requiring continues monitoring and development of new drugs (refs)"
for the previous suggested sentence could you also add the following global reference as well "Vancomycin-Resistant Enterococci: A Review of Antimicrobial Resistance Mechanisms and Perspectives of Human and Animal Health - PubMed (nih.gov)"
line 46-77: this is long unneeded section. i would suggest deleting most of it and only rephrase a nice summary few lines.
line77: please avoid using "we" in your article. i would suggest rephrasing it as follows: "The present report, ................. ........etc"
for the rest of the article follow my suggestion to rewrite this article as suggested int he indicated reference above in the form of a letter report. such modification would require deleting headings, significant reduction and quick major formatting/
a second round of revision is required.
Author Response
Reviewer #1
The authors appreciate the comments and suggestion of the reviewer 1.
- The article is very interesting and adds up needed information. please, reduce the article and present it as a full letter or correspondence following the following recommendations.
The authors have evaluated the possibility to short the paper. Hovewer, correspondence or letter is not included in the special issue chosen by the authors. In addition, the authors believe useful to show all data obtained. Thus, we have shorted the paper erasing the redundant concepts.
- please follow up and add the following article which present very important regional information. Colistin-resistant carbapenemase-producing isolates among Klebsiella spp. and Acinetobacter baumannii in Tripoli, Libya - ScienceDirect
The reference has been added.
- Line 2: i would prefer "analysis" instead of "dissemination" so please modify the title to follow the format over recommendation into letter instead of research article.
The sentence has been corrected.
- line 19: review the following sentence and make it general as or following to the following: "The dramatic increase of infections caused by critically multidrug resistant bacteria is a global health concern"
The sentence has been changed
- line 38: modify "Public Health issue" and no need to capitalize it.
Has been modified.
- line 41: delete "for which new antimicrobial development is urgently needed" and emerge it with the following short sentence into a good fluent sentence like or similar to: "Various bacterial organisms known as ESKAPE (Enterococcus faecium, Staphylococcus aureus, Klebsiella pneumoniae, Acinetobacter baumannii, Pseudomonas aeruginosa and Enterobacter spp.) have emerged as globally critical MDR pathogens requiring continues monitoring and development of new drugs (refs)" for the previous suggested sentence could you also add the following global reference as well "Vancomycin-Resistant Enterococci: A Review of Antimicrobial Resistance Mechanisms and Perspectives of Human and Animal Health - PubMed (nih.gov)"
The sentence has been changed. We have added the suggested reference.
- line 46-77: this is long unneeded section. i would suggest deleting most of it and only rephrase a nice summary few lines.
The sentence has been modified
- line77: please avoid using "we" in your article. i would suggest rephrasing it as follows: "The present report, ................. ........etc"
The sentence has been changed
for the rest of the article follow my suggestion to rewrite this article as suggested int he indicated reference above in the form of a letter report. such modification would require deleting headings, significant reduction and quick major formatting/a second round of revision is required.
The article has been corrected, some parts of the text have been rewritten.
Reviewer 2 Report
This study characterized the antimicrobial resistance profiles and genes of four carbapenemase-producing Enterobacterales strains isolated from a hospitalized patient. The authors tested antimicrobial susceptibility, multilocus sequence typing and ARGs/mobile genetic elements of the four strains. The manuscript is written well. However, I have a few minor suggestions below.
1. Table 1, please refer the full name of “MIC” in the proper place.
2. line 102 and Table 2: please refer the full name of “MLST” and “pMLST” in the proper place.
3. section 2.4, please show the data, like LB plate/agarose gel images, of the conjugation experiments and PCR assays.
4. section 4.1, please provide more details about how the isolates were recovered from the samples.
5. line 226-228, please show the relevant images of the immunochromatographic assay and inhibition test in the supplementary.
Author Response
Reviewer 2#
This study characterized the antimicrobial resistance profiles and genes of four carbapenemase-producing Enterobacterales strains isolated from a hospitalized patient. The authors tested antimicrobial susceptibility, multilocus sequence typing and ARGs/mobile genetic elements of the four strains. The manuscript is written well. However, I have a few minor suggestions below.
- Table 1, please refer the full name of “MIC” in the proper place.
Has been corrected
- line 102 and Table 2: please refer the full name of “MLST” and “pMLST” in the proper place.
Has been corrected
- section 2.4, please show the data, like LB plate/agarose gel images, of the conjugation experiments and PCR assays.
Unfortunately, we have not photographed the conjugation plates and it is very hard to make again a conjugation experiment. We have the images of PCR assays but we believe that is useless to include them.
- section 4.1, please provide more details about how the isolates were recovered from the samples.
Some details concerning the way the isolates have been recovered from the samples.
- line 226-228, please show the relevant images of the immunochromatographic assay and inhibition test in the supplementary.
The images of immunochromatographic assay and inhibition test have been included as Fig. S1 (Supplementary file).
Round 2
Reviewer 1 Report
the revised manuscript looks far better and i thank authors however i have one more note.
line 46: this could be new separate paragraph starting "In the last decade 46 we have witnessed.....etc".
please delete "we have witnessed" and modify the related sentence.
as i explained in my previous reports, lines 46-62" are too long and should be significantly reduced. these lines could be summarized without details.
Author Response
The authors thanks reviewer 1 for the careful revision.
Line 46: the sentence has been changed.
Lines 46-62: this paragraph has been shorted in a single sentence as suggested.